# Abnormal serum levels of liver enzyme markers and related risk factors in type 2 diabetes mellitus patients attending the Buea Regional Hospital, Cameroon

Arnaud Fondjo Kouam [ID][1,2]*, Saturine Mengwe Mofor[1], Madeleine Yvanna Nyangono Essam[2], Armelle Gaelle Kwesseu Fepa[2], Elisabeth Menkem Zeuko'o[1], Armel Jackson Seukep[1], Eléonore Ngounou[1], Pascal Dieudonné Djamen Chuisseu[3], Paul Fewou Moundipa[2], Frédéric Nico Njayou[2]

1 Department of Biomedical Sciences, Faculty of Health Sciences, University of Buea, Buea, Cameroon,
2 Department of Biochemistry, Faculty of Science, University of Yaoundé 1, Yaoundé, Cameroon,
3 Higher Institute of Health Sciences, Université des Montagnes, Bangangté, Cameroon

* arnaudkouam@yahoo.fr, kouam.fondjo@ubuea.cm

## Abstract

The prevalence of type 2 Diabetes Mellitus (T2DM) is increasing globally. Besides the traditional complications associated with T2DM, such as diabetic retinopathy, neuropathy, and kidney diseases, new complications including liver diseases, are increasingly being documented. This study aimed to examine serum biomarkers of liver injury and the related risk factors in T2DM patients at the Buea Regional Hospital, Cameroon. The sociodemographic, clinical, and behavioral characteristics of patients with T2DM were captured using a structured questionnaire. Anthropometric parameters were measured, and the Body Mass Index was calculated. Blood samples were analyzed for biomarkers of liver damage (ALT, AST, GGT, and ALP), considering a liver enzyme profile abnormal if it had more than 2 abnormally elevated values. Bivariate and multivariate logistic regressions analysis were used to identify risk factors, with significance set at P < 0.05. Among the 170 participants recruited, 75.9% were female. The median age was 62 years. Over half (52.9%) were married, 64.7% attended primary school and 55.3% were retired. Also, 59.4% had diabetes for over five years and all reported knowledge of diabetes care. About 73.3% adhered to their medication, 64.7% consumed alcohol, 28.8% smoked tobacco, with 22.4% engaged in physical activity and 77.6% with comorbidities. Blood sugar monitoring was practiced by 80%, with 66.5% having high blood pressure. Healthy weight individuals were 31.2% while 41.2% were obese and 56.5% had abnormal liver enzyme profiles. Five factors: duration of illness, physical inactivity, tobacco smoking, comorbidities and overweight/obesity were significantly (P < 0.05) associated with abnormal liver enzyme profile. Our findings identify risk factors linked to elevated liver enzyme markers indicating liver injury in T2DM patients.

**Data availability statement:** All relevant data are within the manuscript and its Supporting Information files.

**Funding:** This work was supported by the Trimester Research Modernization Funding granted by the Ministry of Higher Education of Cameroon. The funder had no role in study design, data collection and analysis, decision to publish, or preparation of the manuscript.

**Competing interests:** The authors have declared that no competing interests exist.

## Introduction

Diabetes Mellitus (DM) ranks among the most prevalent non-communicable diseases, rising at an alarming pace and affecting a considerable number of individuals. This disease has quickly become a global health issue, mainly due to high calorie intake, lack of physical activity, inadequate diagnosis, and poor management [1,2]. It is now posing a risk of reaching endemic status by 2030, particularly in developing nations [3,4]. Moreover, it is a metabolic disorder characterized by sustained high blood sugar levels with the disturbance of the metabolism of proteins, lipids and carbohydrates as a result of a deficiency in insulin secretion and/or action [5]. Globally, around 1 in 11 adults experience DM, with type 2 DM (T2DM) representing approximately 90% of cases [4]. T2DM arises from dysfunctional pancreatic β-cell activity and an inability to produce enough insulin, coupled with insulin resistance characterized by decreased sensitivity to insulin in target tissues such as muscles, adiposes, and the liver [6].

T2DM negatively affects various systems and organs in the body. Traditional complications well-known to be associated with T2DM are cardiovascular diseases, diabetic kidney disease, retinopathy and peripheral neuropathy [7]. Besides these, emerging complications are continuously reported in diabetic patients with those affecting the liver, which is crucial in the detoxification process, glycaemia regulation and the metabolism of lipids, proteins and carbohydrates [8–10]. Indeed, liver impairment, characterized by abnormally elevated serum levels of liver enzymes such as transaminases: alanine aminotransferase (ALT) and aspartate aminotransferase (AST); alkaline phosphatase (ALP) and γ-glutamyl-transferase (GGT) [11–13], is commonly observed in about 70% of diabetic patients and accounts for roughly 2–4% of fatalities in T2DM [14,15].

Although T2DM represents a significant public health issue in Cameroon, there is a paucity of knowledge regarding the relationship between T2DM and liver injury or dysfunction, which further raises morbidity and mortality among Cameroonian diabetic patients. Accordingly, continuous monitoring of serum liver enzymes and identification of potential risk factors associated with their abnormal levels in T2DM patients are necessary to detect early signs of hepatic injury. This may prompt medical intervention to prevent further detrimental complications for the patients [16]. Therefore, the purpose of this study was to assess the serum levels of liver enzyme markers and identify the potential risk factors related to their abnormal levels in T2DM patients attending the Buea Regional Hospital, South-West Region, Cameroon.

## Materials and methods

### Study design and settings

This descriptive cross-sectional study followed by laboratory analysis took place at the Diabetes Unit of the Buea Regional Hospital, from June to September 2024. The city of Buea serves as the Headquarters of the South-West Region of Cameroon. It is located on the eastern slope of Mount Cameroon. It has coordinates 4°10'0″N994'0″E ╱4.16667°N9.23333°E with an elevation of 896 m above sea level. The Buea Regional

Hospital is classified as a secondary health institution, that delivers both outpatient and inpatient services for various diseases, including chronic conditions such as hypertension, stroke, kidney and cardiac diseases, and DM.

## Target population, sampling technique and sample size estimation

The target population was diabetic patients aged over 21 years. The study's inclusion criteria consisted of diabetic individuals diagnosed with T2DM, T2DM patients who have been on antidiabetic medication for no less than 6 months and T2DM patients who willingly and freely consented to participate in the study by signing the informed consent form. Exclusion criteria included diabetic patients with a history of liver diseases, clinical symptoms of acute hepatitis, infected with hepatitis C or B virus, or taking hepatotoxic drugs. A convenient random sampling technique was used for participant recruitment. After approaching a potential participant, the goals of the investigation, as well as the potential risks and benefits, were clearly explained and only those who gave their consent by signing the informed consent form were enrolled in the study. The estimated sample size was calculated using the Lorentz formula (1).

$$N = \frac{Z^2 \times p(1-p)}{d^2}$$

(1)

Where: $N$ is the calculated sample size; $Z = 1.96$ is the typical value of the degree of confidence at 95%; $p = 7.1\%$ is the prevalence of diabetes among adults living in Cameroon [17]. $d = 5\%$ is the margin error permitted.

After calculation, $N \approx 102$. For a better representation of the population, 20% of $N$ was added. Therefore, the minimum number of diabetic patients to enroll in this study was estimated at 123 participants.

## Ethical consideration

The study was conducted in accordance with guidelines from the declarations of Helsinki. Ethical Clearance for this study was obtained from the Institutional Review Board of the Faculty of Health Sciences, University of Buea (Ref N°: 2024/2484-03/UB/SG/IRB/FHS). In addition, administrative authorizations were issued by the Regional Health Office of the South-West Region, Ministry of Public Health, Cameroon (Ref N°: P42/MINSANTE/SWR/RDPH/CB.PT/183/293) and from the director of the Buea Regional Hospital (Ref N°: 22/05/2024/MPH/SWRDPH/BRH/IRB), respectively. Only patients who provided their consent by signing the written informed consent form were admitted in the study. The information collected during the interview was not disclosed and the participant's names were replaced with codes during the data collection.

## Data collection procedure

**General characteristics of diabetic patients enrolled in the study.** An overview of the characteristics of diabetic patients recruited for this investigation was captured through a structured questionnaire (Additional information S1 File) administered in person. The questionnaire allowed for gathering information regarding socio-demographic data, the history and monitoring of illness, eating habits and physical activity practices, alcohol consumption, and the use of tobacco products. In addition, blood pressure and anthropometric parameters including height and weight were measured, and the Body Mass Index (BMI) was subsequently calculated; each participant was classified as underweight, normal weight, overweight, or obese based on a BMI of <18.5, [18.5–24.9], [25–29.9], and >30, respectively. At the end of the interview, 5 mL of blood was collected via venipuncture into a tube free of anticoagulant and centrifuged (3000 × g, 10 min, 4°C). The serum obtained was used to measure liver enzyme markers.

**Evaluation of some biochemical markers of liver injury in diabetic patients.** The biochemical markers of liver injury were assessed by measuring the serum activity of some liver enzymes, including alanine aminotransferase (ALT), aspartate aminotransferase (AST), alkaline phosphatase (ALP), and γ-glutamyl-transferase (GGT) using commercial

assay kits (Catalog N° REF_80227, Catalog N° REF_80225, Catalog N° REF_92314 and Catalog N° REF_81310 respectively for ALT, AST, ALP, and GGT) purchased from BIOLABO, Les Hautes Rives, Maizy, France. The assays were performed in accordance with the manufacturer's instructions using a semi-automatic spectrophotometer (Semi-auto Chemistry Analyzer, BIOBASE-Claire, BioBase, Jinan, Shandong, China).

Briefly, the kit used to measure serum ALT activity included one reagent R1 containing L-Alanine (500 mM), 2-oxoglutarate (15 mM) and TRIS (100 mM, pH 7.8), NADH (0.18 mM), and Lactate dehydrogenase (1600 U/L). In 1 cm path length thermostated cuvette at 37°C, 1000 µL of R1 reagent was mixed with 100 µL of serum sample. The initial absorbance was recorded after 60 seconds at 340 nm, followed by further recording of the absorbance every minute for 3 minutes, and the activity of ALT in the serum sample was automatically deduced by the pre-calibrated spectrophotometer.

Regarding AST Assay, the kit included one reagent R1 containing L-Aspartate (200 mM), 2-oxoglutarate (12 mM) and TRIS (80 mM, pH 7.8), EDTA (5 mM), NADH (0.18 mM), Malate dehydrogenase (495 U/L), and Lactate dehydrogenase (820 U/L). In 1 cm path length thermostated cuvette at 37°C, 1000 µL of R1 reagent was mixed with 100 µL of serum sample. The initial absorbance was recorded after 60 seconds at 340 nm, followed by further recording of the absorbance every minute for 3 minutes, and the activity of AST in the serum sample was automatically deduced by the pre-calibrated spectrophotometer.

Concerning ALP activity, the kit includes two reagents: reagent R1 (Alkaline phosphate buffer) containing Diethanol-amine buffer (1 M, pH 10), and reagent R2 (Alkaline phosphate substrate) containing para-nitrophenyl phosphate (10 mM). The working reagent was prepared by mixing the content of vial R1 with R2. Then, in 1 cm path length thermostated cuvette at 37°C, 1000 µL of working reagent was mixed with 10 µL of serum sample. After 1 minute at 405 nm, the absorbance was recorded every minute for 3 minutes, and the activity of ALP in the serum sample was automatically deduced by the pre-calibrated spectrophotometer.

The kit used for the measurement of GGT activity included two reagents: reagent R1 (Gamma GT Buffer) containing Glycylglycine (100 mM) and TRIS buffer (95 mM, pH 8.25), and reagent R2 Gamma GT substrate) containing L-G-glutamyl-3-carboxy-4-nitroanilide (80 mM). To prepare the working reagent, 10 mL R1 vial were measured and transferred promptly into R2, then the mixture of R2 vial was transferred into R1 vial. In 1 cm path length thermostated cuvette at 37°C, 1000 µL of working reagent was mixed with 50 µL of serum sample. After 30 second at 405 nm, the absorbance was recorded every minute for 3 minutes, and the activity of GGT in the serum sample was automatically deduced by the pre-calibrated spectrophotometer.

**Estimation of the alteration of serum liver enzyme markers.** The level of alteration of the serum liver enzyme activity was estimated by considering the normal range of values (reference values) for each enzyme assessed. These reference values were: from 10 to 42 IU/L, 8–39 IU/L, 40–129 IU/L and 11–50 IU/L respectively for ALT, AST, ALP and GGT, as indicated in the manufacturer's instructions. Accordingly, any value found within its normal range or higher than upper limit of the normal range of values (ULN) was considered normal or high, respectively. Similarly, any patient with more than 2 high values was considered to have an abnormal status of liver enzyme profile.

## Data management and statistical analysis

After checking that all sections of the questionnaire had been completed, the data collected and the results of the laboratory analyses for each participant were saved in Excel 2013 (Microsoft Corporation, USA) (Additional information S2 File), and then exported to the statistical analysis software SPSS (Statistical Package for Social Sciences) version 25.0 (SPSS Inc., USA) or GraphPad Prism version 8.0.2. Descriptive statistics were performed using SPSS software. Qualitative variables were presented as frequency and percentage (%). Quantitative variables were first tested for normality using the Kolmogorov-Smirnov test. Variables that followed a normal distribution and those that did not pass the test were expressed as mean ± standard deviation or median and interquartile range respectively. Comparison of median values between two categories was done by the non-parametric Mann Whitney $U$ test. Risk factors associated with the abnormal

status of liver enzyme profile were determined through bivariate and multivariate logistic regression analysis. The significance threshold was declared at $P < 0.05$.

## Results

### Socio-demographic characteristics of the enrolled T2DM patients

A total of 170 patients were recruited for the study. The gender ratio was 3.15, favoring females, who made up 75.9% (129/170) of the participants. The median age of the participants was 62 years, with an interquartile range of 55–70 years for the 25% and 75% percentiles, respectively. A significant portion of the participants, 66.5% (113/170), were over 60 years old. Additionally, more than half of the participants were married (52.9%; 90/170), and 64.7% (110/170) attended primary school. In terms of occupation, 55.3% (94/170) of the enrolled patients were retired (Table 1).

### Behavioral and clinical features of study participants

Table 2 outlines the frequency distribution of study participants based on their behavioral and clinical characteristics. Regarding the duration of illness, 59.4% (101/170) reported being diagnosed with diabetes for over five years. All participants (100%; 170/170) indicated they had knowledge of diabetes care, while 73.3% (128/170) strictly adhered to their anti-diabetic medications. Among the participants, 64.7% (110/170) consumed alcohol, and 28.8% (49/170) were tobacco smokers. Only 22.4% (38/170) of the enrolled diabetic patients engaged in physical activity and 77.6% (132/170) had at least one comorbidity. Systematic blood sugar monitoring was reported by 80% (136/170) of participants, while 66.5% (113/170) had high blood pressure. Based on their BMI index, those with healthy weight comprised 31.2% (33/170), whereas 27.6% (47/170) were overweight and 41.2% (70 out of 170) were obese. Abnormally high levels of serum ALT, AST, ALP and GGT activities were observed in 61.8% (105/170), 62.4% (106/170), 37.3% (64/170) and 50% (85/170) of

**Table 1. Socio-demographic characteristics of diabetic patients attending the Buea Regional Hospital.**

| Socio-demographic characteristics | Categories | Frequency (n) | Percentage (%) |
|---|---|---|---|
| **Gender** | Female | 129 | 75.9 |
| | Male | 41 | 24.1 |
| | **Total** | **170** | **100.0** |
| **Age group** | [21-40[ | 6 | 3.5 |
| | [40-60] | 51 | 30.0 |
| | >60 | 113 | 66.5 |
| | **Total** | **170** | **100.0** |
| **Marital status** | Married | 90 | 52.9 |
| | Single | 17 | 10.0 |
| | Widow(er) | 63 | 37.1 |
| | **Total** | **170** | **100.0** |
| **Education** | Primary | 110 | 64.7 |
| | Secondary | 36 | 21.2 |
| | University | 24 | 14.1 |
| | **Total** | **170** | **100.0** |
| **Occupation** | Civil servant | 14 | 8.2 |
| | Employee | 62 | 36.5 |
| | Retired | 94 | 55.3 |
| | **Total** | **170** | **100.0** |

**Table 2. Behavioral and clinical features of diabetic patients attending the Buea Regional Hospital.**

| Behavioral/ Clinical features | Categories | Frequency (n) | Percentage (%) |
|---|---|---|---|
| **Duration of illness** | ≤ 5 years | 69 | 40.6 |
| | > 5 years | 101 | 59.4 |
| | **Total** | **170** | **100.0** |
| **Knowledge of diabetes care** | Yes | 170 | 100.0 |
| | No | 0 | 0 |
| | **Total** | **170** | **100.0** |
| **Adherence to antidiabetic drugs** | Yes | 128 | 73.3 |
| | No | 42 | 24.7 |
| | **Total** | **170** | **100.0** |
| **Alcohol consumption** | Yes | 110 | 64.7 |
| | No | 60 | 35.3 |
| | **Total** | **170** | **100.0** |
| **Tobacco smoking** | Yes | 49 | 28.8 |
| | No | 121 | 71.2 |
| | **Total** | **170** | **100.0** |
| **Practice of physical activity** | Yes | 38 | 22.4 |
| | No | 132 | 77.6 |
| | **Total** | **170** | **100.0** |
| **Comorbidity** | Yes | 132 | 77.6 |
| | No | 38 | 22.4 |
| | **Total** | **170** | **100.0** |
| **Blood sugar monitoring** | Yes | 136 | 80.0 |
| | No | 34 | 20.0 |
| | **Total** | **170** | **100.0** |
| **Level of blood pressure** | High | 113 | 66.5 |
| | Normal | 57 | 33.5 |
| | **Total** | **170** | **100.0** |
| **Interpretation of BMI** | Healthy weight | 53 | 31.2 |
| | Overweight | 47 | 27.6 |
| | Obese | 70 | 41.2 |
| | **Total** | **170** | **100.0** |
| **ALT (IU/L)** | High> (42) | 105 | 61.8 |
| | Normal <(42) | 65 | 38.2 |
| | **Total** | **170** | **100.0** |
| **AST (IU/L)** | High> (39) | 106 | 62.4 |
| | Normal <(39) | 64 | 37.6 |
| | **Total** | **170** | **100.0** |
| **ALP (IU/L)** | High> (129) | 64 | 37.6 |
| | Normal <(129) | 106 | 62.4 |
| | **Total** | **170** | **100.0** |
| **GGT (IU/L)** | High> (50) | 85 | 50.0 |
| | Normal <(50) | 85 | 50.0 |
| | **Total** | **170** | **100.0** |
| **Status of liver enzyme profile** | Abnormal | 96 | 56.5 |
| | Normal | 74 | 43.5 |
| | **Total** | **170** | **100.0** |

BMI: Body Mass Index; ALT: Alanine amino transferase; AST: Aspartate aminotransferase; ALP: Alkaline phosphatase; GGT: γ-glutamyl-transferase. Comorbidity includes at least one of the following conditions: High Blood Pressure, Kidney Diseases, Cardiovascular diseases, Diabetes neuropathy and retinopathy.

study participants, respectively. Consequently, the prevalence of abnormal liver enzyme profiles among the enrolled T2DM patients was estimated to be 56.5% (96/170).

### Association between the socio-demographic characteristics and the status of liver enzyme profile

The potential socio-demographic factors associated with abnormal liver enzyme profiles were evaluated using bivariate logistic regression analysis as summarized in Table 3.

Although a higher percentage of females (45.9%; 78/129) were affected by abnormal liver enzyme profiles compared to males (10.6%; 18/ 41), no significant association was found between gender and liver enzyme status (cOR: 1.95; CI: 0.96–3.97; P = 0.065). Additionally, the analyses indicated that age group (cOR: 0.69 and 0.71; CI: 0.13–3.55 and 0.37–1.39; P = 0.654 and 0.321), marital status (cOR: 0.72 and 1.20; CI: 0.37–1.38 and 0.39–3.68; P = 0.322 and 0.742), education level (cOR: 0.94 and 1.92; CI: 0.39–2.29 and 0.66–3.97; P = 0.898 and 0.232), and occupation (cOR: 1.12 and 0.54; CI: 0.35–3.60 and 0.96–3.97; P = 0.853 and 0.067) did not significantly influence liver enzyme status among the enrolled T2DM patients.

### Relationship between the behavioral and clinical factors and the status of liver enzyme profile of study participants

Table 4 summarizes the statistical associations between the behavioral and clinical factors related to an abnormal liver enzyme profile, determined through bivariate and multivariate logistic regression analysis, as indicated below.

**Possible factors influencing the abnormal status of liver enzyme profile.** For the bivariate analysis, a simple logistic regression model was used at 95% confidence interval (CI) with a cut-off point p-value set at 0.05 to identify factors for multivariate analysis. The clinical and behavioral factors identified as significantly influencing the liver enzyme status were as follows: duration of illness (cOR: 3.73; CI: 1.96–7.12; P < 0.0001); alcohol consumption (cOR: 0.39; CI: 0.21–0.75; P = 0.004); tobacco smoking (cOR: 0.029; CI: 0.007–0.12; P < 0.0001); comorbidity (cOR: 0.31; CI: 0.14–0.65; P = 0.002); practice of physical activity (cOR: 7.50; CI: 3.17–17.72; P < 0.0001); and BMI index (cOR: 0.12; CI: 0.048–0.30; P < 0.0001).

**Factors associated with abnormal liver enzyme profile.** Following the bivariate analysis, a multivariate logistic regression analysis was performed in the same conditions to determine the factors associated with abnormal liver enzyme profile among those identified as influencing the liver enzyme status. Five factors were found to be significantly associated with abnormal levels of liver enzyme among the enrolled T2DM patients. These factors were: duration of illness (aOR: 6.23; CI: 2.04–19.08; P = 0.001); tobacco smoking (aOR: 0.022; CI: 0.003–0.14; P < 0.0001); practice of physical activity (aOR: 4.64; CI: 1.34–16.07; P = 0. 015); comorbidity (aOR: 0.18; CI: 0.049–0.70; P = 0.013); and BMI (aOR: 0.048; CI: 0.011–0.21; P < 0.0001). This means that the enrolled T2DM participants with a duration of illness greater than 5 years (aOR: 6.23) and those not engaged in physical activity (aOR: 4.64) were significantly (P < 0.05) at higher risk of presenting abnormal liver enzyme status, compared to the T2DM patients with a duration of illness less than 5 years and those who engaged in physical activity, respectively. Similarly, T2DM patients who did not smoke (aOR: 0.022), without comorbidity (aOR: 0.18), and with healthy weight (aOR: 0.048) were significantly (P < 0.05) less at risk of having an abnormal liver enzyme profile, compared to the patients who smoked, had comorbidity and those who were overweight or obese, respectively.

### Influence of the identified risk factors on the serum levels of liver enzyme activity

Fig 1 depicts the variation in liver enzyme (ALT, AST, GGT, and ALP) activity according to the risk factors associated with their abnormal profile. The median values of serum ALT, AST, and GGT activities were significantly (P < 0.05) elevated among T2DM patients with a duration of illness greater than 5 years (Fig 1A), those who smoked (Fig 1B), those who

were physically inactive (Fig 1C), those with comorbidities (Fig 1D), and those who were obese or overweight (Fig 1E), compared to those with a duration of illness less than 5 years, non-smokers, physically active individuals, those without comorbidities and those with a healthy weight, respectively. Regarding the median values of ALP, no significant (P > 0.05) difference was observed between patients with more than 5 years of illness duration and those with less than 5 years (Fig 1A), or between healthy weight and overweight patients (Fig 1E). In contrast, median values of ALP activity were significantly increased in T2DM patients who smoked (Fig 1B), were physically inactive (Fig 1C), had comorbidities (Fig 1D), and were obese (Fig 1E), compared to non-smokers, physically active individuals, those without comorbidities and healthy weight patients, respectively.

## Discussion

The rising global prevalence of Type 2 Diabetes Mellitus (T2DM) poses significant health challenges and complications. In addition to the traditional complications associated with T2DM, such as diabetic retinopathy, neuropathy, and kidney diseases, emerging complications including liver diseases are increasingly being reported. This trend is especially evident in low- to middle-income countries like Cameroon, where healthcare resources are often limited [14,18,19]. This study examined the serum levels of biomarkers of liver injury in T2DM patients attending the Buea Regional Hospital of Cameroon, provides valuable insights into the interaction between liver health, clinical and behavioral features of this target population. Indeed, identifying the risk factors that contribute to abnormal liver enzyme levels can facilitate the development of effective intervention strategies to prevent the occurrence of liver diseases in T2DM patients. Hence, several clinical and behavioral factors related to abnormal serum liver enzyme levels were identified. These included the duration of illness,

**Table 3. Bivariate logistic regression analysis for the association between the socio-demographic characteristics of diabetic patients and their status of liver enzyme profile.**

| Variables | Categories | Status of liver enzyme profile | | | Bivariate logistic regression | | |
|---|---|---|---|---|---|---|---|
| | | Abnormal n, (%) | Normal n, (%) | Total | cOR | [95% CI] | P value |
| **Gender** | Female | 78 (45.9) | 51 (30.0) | **129 (75.9)** | 1.95 | [0.96 − 3.97] | 0.065 |
| | Male | 18 (10.6) | 23 (13.5) | **41 (24.1)** | 1 | / | / |
| | **Total** | **96 (56.5)** | **74 (43.5)** | **170 (100.0)** | / | / | / |
| **Age group** | [21-40[ | 3 (1.8) | 3 (1.8) | **6 (3.5)** | 0.69 | [0.13 − 3.55] | 0.654 |
| | [40-60] | 26 (15.3) | 25 (14.7) | **51 (30.0)** | 0.71 | [0.37 − 1.39] | 0.321 |
| | >60 | 67 (39.4) | 46 (27.1) | **113 (65.5)** | 1 | / | / |
| | **Total** | **96 (56.5)** | **74 (43.5)** | **170 (100.0)** | / | / | / |
| **Marital status** | Married | 47 (27.6) | 43 (25.3) | **90 (52.9)** | 0.72 | [0.37 − 1.38] | 0.322 |
| | Single | 11 (6.5) | 6 (3.5) | **17 (10.0)** | 1.20 | [0.39 − 3.68] | 0.742 |
| | Widow(er) | 38 (22.4) | 25 (14.7) | **63 (37.1)** | 1 | / | / |
| | **Total** | **96 (56.5)** | **74 (43.5)** | **170 (100.0)** | / | / | / |
| **Education** | Primary | 58 (34.1) | 52 (30.6) | **110 (64.7)** | 0.94 | [0.39 − 2.29] | 0.898 |
| | Secondary | 25 (14.7) | 11 (6.5) | **36 (21.2)** | 1.92 | [0.66 − 5.61] | 0.232 |
| | University | 13 (7.6) | 11 (6.5) | **24 (14.1)** | 1 | / | / |
| | **Total** | **96 (56.5)** | **74 (43.5)** | **170 (100.0)** | / | / | / |
| **Occupation** | Civil servant | 9 (5.3) | 5 (2.9) | **14 (8.2)** | 1.12 | [0.35 − 3.60] | 0.853 |
| | Employee | 29 (17.1) | 33 (19.4) | **62 (36.5)** | 0.54 | [0.96 − 3.97] | 0.067 |
| | Retired | 58 (34.1) | 36 (21.2) | **94 (55.3)** | 1 | / | / |
| | **Total** | **96 (56.5)** | **74 (43.5)** | **170 (100.0)** | / | / | / |

cOR: Crude Odd Ratio; CI: Confidence Interval.

**Table 4. Bivariate and multivariate logistic regression analysis for the behavioral and clinical factors associated with the abnormal liver enzyme profile in diabetic patients attending the Buea Regional Hospital.**

| Behavioral/ Clinical features | Categories | Status of liver enzyme profile | | | Bivariate logistic regression | | | Multivariate logistic regression | | |
|---|---|---|---|---|---|---|---|---|---|---|
| | | Abnormal n, (%) | Normal n, (%) | Total | cOR | [95% CI] | P values | aOR | [95% CI] | P values |
| Duration of illness | > 5 years | 70 (41.2) | 31 (18.2) | **101 (59.4)** | **3.73** | [1.96 – 7.12] | **<0.0001*** | **6.23** | [2.04 – 19.08] | **0.001*** |
| | ≤ 5 years | 26 (15.3) | 43 (25.3) | **69 (40.6)** | 1 | / | / | 1 | / | / |
| | Total | **96 (56.5)** | **74 (43.5)** | **170 (100.0)** | / | / | / | / | / | / |
| Adherence to antidiabetic drugs | No | 22 (12.9) | 20 (11.8) | **42 (24.7)** | 0.80 | [0.39 – 1.61] | 0.538 | / | | |
| | Yes | 74 (43.5) | 54 (31.8) | **128 (75.3)** | 1 | / | / | | | |
| | Total | **96 (56.5)** | **74 (43.5)** | **170 (100.0)** | / | / | / | | | |
| Alcohol consumption | No | 25 (14.7) | 35 (20.6) | **60 (35.3)** | **0.39** | [0.21 – 0.75] | **0.004*** | 0.81 | [0.27 – 2.40] | 0.703 |
| | Yes | 71 (41.8) | 39 (22.9) | **110 (64.7)** | 1 | / | / | 1 | / | / |
| | Total | **96 (56.5)** | **74 (43.5)** | **170 (100.0)** | / | / | / | / | / | / |
| Tobacco smoking | No | 49 (28.8) | 72 (42.4) | **121 (71.2)** | **0.029** | [0.007 – 0.12] | **<0.0001*** | **0.022** | [0.003– 0.14] | **<0.0001** |
| | Yes | 47 (27.6) | 2 (1.2) | **49 (28.8)** | 1 | / | / | 1 | / | / |
| | Total | **96 (56.5)** | **74 (43.5)** | **170 (100.0)** | / | / | / | / | / | / |
| Practice of physical activity | No | 88 (51.8) | 44 (25.9) | **132 (77.6)** | **7.50** | [3.17 – 17.72] | **<0.0001*** | **4.64** | [1.34 – 16.07] | **0.015*** |
| | Yes | 8 (4.7) | 30 (17.6) | **38 (22.4)** | 1 | / | / | 1 | | |
| | Total | **96 (56.5)** | **74 (43.5)** | **170 (100.0)** | / | / | / | | | |
| Comorbidity | No | 13 (7.6) | 25 (14.7) | **38 (22.4)** | **0.31** | [0.14 – 0.65] | **0.002*** | **0.18** | [0.049 – 0.70] | **0.013*** |
| | Yes | 83 (48.8) | 49 (28.8) | **132 (77.6)** | 1 | / | / | 1 | / | / |
| | Total | **96 (56.5)** | **74 (43.5)** | **170 (100.0)** | / | / | / | / | / | / |
| Blood sugar monitoring | No | 18 (10.6) | 16 (9.4) | **34 (20.0)** | 0.84 | [0.39 – 1.78] | 0.643 | / | | |
| | Yes | 78 (45.9) | 58 (34.1) | **136 (80.0)** | 1 | / | / | | | |
| | Total | **96 (56.5)** | **74 (43.5)** | **170 (100.0)** | / | / | / | | | |
| Level of blood pressure | High | 67 (39.4) | 48 (27.1) | 113 (66.5) | 1.40 | [0.74 – 2.67] | 0.297 | / | | |
| | Normal | 29 (17.1) | 28 (16.5) | 57 (33.5) | 1 | / | / | | | |
| | Total | **96 (56.5)** | **74 (43.5)** | **170 (100.0)** | / | / | / | | | |
| Interpretation of BMI | Healthy weight | 10 (5.9) | 43 (25.3) | 53 (31.2) | **0.12** | [0.048 – 0.30] | **0.0001*** | **0.048** | [0.011 – 0.21] | **<0.0001*** |
| | Obese | 55 (32.4) | 15 (8.8) | 70 (41.2) | 1.89 | [0.82 – 4.34] | 0.132 | 2.55 | [0.80 – 8.09] | 0.112 |
| | Overweight | 31 (18.2) | 16 (9.4) | (47 (27.6) | 1 | / | / | 1 | / | / |
| | Total | **96 (56.5)** | **74 (43.5)** | **170 (100.0)** | / | / | / | / | / | / |

BMI: Body Mass Index; cOR: Crude Odd Ratio; aOR: Adjusted Odd Ratio; CI: Confidence Interval; The bold cOR or aOR and P-value are indicators of a significant association.

practice of physical activity, tobacco smoking, presence of comorbidities such as High Blood Pressure, Kidney Diseases, Cardiovascular diseases, Diabetes neuropathy and retinopathy and high Body Mass Index (BMI). Each of these factors is well known to affect liver health [20–22] and addressing them holistically may improve health outcomes among T2DM patients.

The duration of T2DM is a fundamental factor influencing the development of liver-related complications. Indeed, oxidative injury and the abnormal inflammatory response as a results of persistent hyperglycemia can lead to liver injury, non-alcoholic fatty liver disease and steatosis, which can further progress to liver fibrosis and cirrhosis [10]. In addition, insulin resistance promotes de novo lipogenesis and increases uptake of fatty acids into liver cells, whose consequences

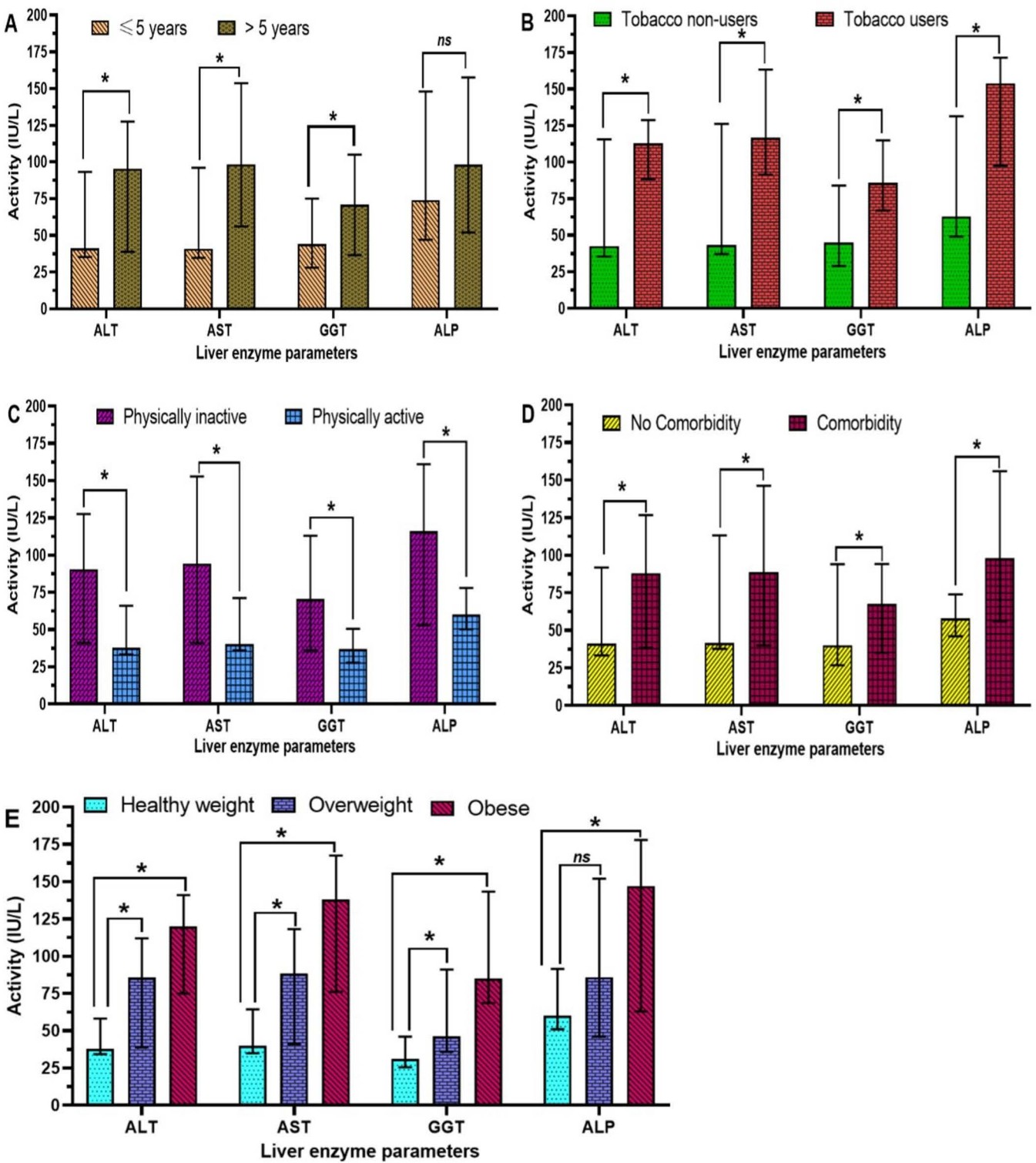

**Fig 1. Variation of liver enzyme activity according to the risk factors associated with their abnormal levels.** *(A) Effect of duration of illness; (B): Effect of tobacco smoking; (C): Effect of physical activity; (D): Effect of comorbidities; (E): Effect of Body Mass Index. Comparison of median values between two categories was done by the non-parametric Mann Whitney U test. * Values significantly different (P<0.05);* **ns** *Values non-significantly different (P>0.05).*

are the increase production of reactive oxygen species (ROS) and pro-inflammatory cytokines which mediate liver injuries [23,24]. Patients with a longer duration of diabetes are at an elevated risk of accumulating lipids into hepatic tissues, leading to inflammation and in severe cases, may progress to non-alcoholic steatohepatitis (NASH) and ultimately result in liver cirrhosis or hepatocellular carcinoma [10,25]. This study showed that T2DM patients with a duration of illness greater than 5 years display a significant (P<0.05) increased serum level of ALT, AST, GGT and ALP activities and were significantly more at risk (aOR: 6.238; P=0.001) of having abnormal liver enzyme profile, compared to those with less than 5 years history of the disease. These observations suggest a possible development of liver pathology in these old T2DM patients, which need to be confirmed by further analysis. It is important for healthcare providers to implement systematic screenings for liver function markers, especially in patients with a prolonged history of diabetes to reduce the risks related to the duration of the disease. This could help identify early signs of liver damage, allowing for timely interventions that could prevent the progression of liver disease. Also, educational programs aimed at raising awareness among T2DM patients about the potential hepatic complications related to the duration of their illness can empower them to take an active role in their disease management.

Moreso, regular physical activity is universally recognized for its beneficial effects on metabolic function, especially in managing diabetes [26,27]. Our findings indicate that patients who engaged in regular physical activity exhibited significant (P<0.05) lower serum liver enzyme levels and were significantly less at risk of presenting abnormal status of liver enzyme profile, compared to physically inactive patients (aOR: 4.64; P=0. 015). Given that physical activity is known to enhance lipid profiles and reduce systemic inflammation, creating a healthier environment for hepatic function is important, these findings can be linked to beneficial effect of physical activity on lipid metabolism and improved insulin sensitivity, as reported by Cannata et al. [28]. This study also emphasizes that incorporating structured exercise regimen into diabetes management plans cannot be overstated. Accordingly, healthcare providers should consider implementing tailored exercise programs for their patients, encouraging activities that are accessible and enjoyable to foster long-term adherence. In addition, engaging patients in community-focused activities may also promote social support systems, which can further motivate sustained participation of T2DM patients in physical activities.

The detrimental effects of tobacco smoking on health are well established and the study indicates that smoking significantly correlates with elevated liver enzyme levels in T2DM patients [29,30]. Tobacco use has been associated with an increased risk of liver disease through mechanisms involving oxidative stress, inflammation and metabolic dysregulation [30,31]. The chemicals such cadmium, present in tobacco smoke can induce hepatic oxidative injury, characterized by abnormal increased serum levels of transaminases [32]. In this study, up to 28.8% of the enrolled T2DM patients were tobacco users. These patients presented significantly (P<0.05) increased serum levels of ALT, AST, GGT, and ALP and were at significantly (aOR: 0.022; P<0.0001) higher risk of displaying abnormal liver enzyme profile, compared to non-smoker patients. These observations suggest that smoking can promote early development of liver complications in T2DM patients. Accordingly, healthcare interventions should prioritize smoking cessation programs for T2DM patients by providing resources for cessation support, such as counseling and pharmacotherapy. Also, creating awareness campaigns that educate patients on the specific risks associated with smoking and liver health can empower individuals to make informed lifestyle choices.

The presence of comorbidities, such as hypertension, obesity, cardiovascular and kidney diseases, and dyslipidemia, represents further challenges in managing the health of T2DM patients [33,34]. In this study, 77.6% (132/170) of participants presented at least one of these comorbid conditions and were more likely to display abnormal liver enzyme levels, when compared to patients without any comorbidity. In fact, multivariate logistic regression analysis showed that comorbidities were significantly (aOR: 0.18; P=0.013) associated with abnormal status of liver enzyme profile. These findings highlight that effective diabetes treatment must include strategies to address all comorbid conditions simultaneously. This may include coordinated efforts among healthcare providers to ensure a cohesive treatment plan that considers managing not only glycaemia, but also blood pressure, and lipidemia. This could be helpful for better management of T2DM and its

related complications, including liver damage. This study also reveals a significant (aOR: 0.048; P < 0.0001) association between elevated BMI and abnormal serum levels of liver enzyme markers in T2DM patients. Indeed, individuals with a higher BMI, particularly those categorized as overweight or obese, are more likely to exhibit elevated liver enzymes, which may indicate liver stress or damage [35–37]. These observations suggest that maintaining a healthy weight could potentially reduce liver complications in diabetic patients, emphasizing the importance of weight management in this population.

This study has some limitations. First, the sample size was relatively small and was limited to a single health facility, the Regional Hospital of Buea, Cameroon. The findings should be validated through further research involving other healthcare facilities across Cameroon and larger participant cohorts. Such studies would help establish trends in liver injury among diabetic patients in the country. Additionally, the study only examined liver injury markers and did not include other liver function tests, such as total bilirubin, conjugated bilirubin, and unconjugated bilirubin. Future investigations are therefore needed to assess both liver damage and various liver function parameters to gain a more comprehensive understanding of the hepatic condition of diabetic patients.

## Conclusion

In summary, the findings from the present study which aimed to measure the serum levels of liver enzyme and identify the potential risk factors related to their abnormal levels in T2DM patients attending the Buea Regional Hospital-Cameroon, several clinical and behavioral factors, including the duration of illness, non-practice of physical activity, tobacco smoking, presence of comorbidities, and elevated BMI values were strongly associated with abnormal serum levels of liver enzyme, suggesting the progressive occurrence of hepatic damage. These observations pave the way for future research that will validate these associations in larger cohorts and explore the underlying mechanisms linking these risk factors to hepatic injury.

## Supporting information

**S1 File. Questionnaire.**
(DOCX)

**S2 File. Research Data.**
(XLSX)

## Acknowledgments

The authors express their gratitude to all participants involved in this study and the laboratory staff of the Buea Regional Hospital for their technical assistance.

## Author contributions

**Conceptualization:** Arnaud Fondjo Kouam, Paul Fewou Moundipa, Frédéric Nico Njayou.

**Data curation:** Arnaud Fondjo Kouam, Saturine Mengwe Mofor, Madeleine Yvanna Nyangono Essam, Armel Jackson Seukep, Pascal Dieudonné Djamen Chuisseu.

**Formal analysis:** Saturine Mengwe Mofor, Madeleine Yvanna Nyangono Essam, Armelle Gaelle Kwesseu Fepa, Elisabeth Menkem Zeuko'o, Armel Jackson Seukep, Eléonore Ngounou, Pascal Dieudonné Djamen Chuisseu, Frédéric Nico Njayou.

**Investigation:** Arnaud Fondjo Kouam, Saturine Mengwe Mofor, Madeleine Yvanna Nyangono Essam.

**Methodology:** Arnaud Fondjo Kouam, Saturine Mengwe Mofor.

**Resources:** Arnaud Fondjo Kouam, Saturine Mengwe Mofor, Armelle Gaelle Kwesseu Fepa, Eléonore Ngounou, Paul Fewou Moundipa, Frédéric Nico Njayou.

**Software:** Arnaud Fondjo Kouam, Madeleine Yvanna Nyangono Essam.

**Supervision:** Arnaud Fondjo Kouam, Madeleine Yvanna Nyangono Essam, Elisabeth Menkem Zeuko'o, Armel Jackson Seukep, Eléonore Ngounou, Paul Fewou Moundipa, Frédéric Nico Njayou.

**Validation:** Arnaud Fondjo Kouam, Armelle Gaelle Kwesseu Fepa, Elisabeth Menkem Zeuko'o, Pascal Dieudonné Djamen Chuisseu, Frédéric Nico Njayou.

**Visualization:** Arnaud Fondjo Kouam, Madeleine Yvanna Nyangono Essam, Elisabeth Menkem Zeuko'o, Eléonore Ngounou, Pascal Dieudonné Djamen Chuisseu, Paul Fewou Moundipa, Frédéric Nico Njayou.

**Writing – original draft:** Arnaud Fondjo Kouam, Saturine Mengwe Mofor, Madeleine Yvanna Nyangono Essam.

**Writing – review & editing:** Arnaud Fondjo Kouam, Armelle Gaelle Kwesseu Fepa, Elisabeth Menkem Zeuko'o, Armel Jackson Seukep, Eléonore Ngounou, Pascal Dieudonné Djamen Chuisseu, Paul Fewou Moundipa, Frédéric Nico Njayou.

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
