## [Decision Letter · Decision Letter 0]

Dear Dr. KOUAM,

**Additionally, please consult with a statistician about the analysis you have conducted, and have an native speaker edited the English writing. **

We look forward to receiving your revised manuscript.

Kind regards,

Suyan Tian

Academic Editor

PLOS ONE

**Journal Requirements:**

1. When submitting your revision, we need you to address these additional requirements. Please ensure that your manuscript meets PLOS ONE's style requirements, including those for file naming. The PLOS ONE style templates can be found at https://journals.plos.org/plosone/s/file?id=wjVg/PLOSOne_formatting_sample_main_body.pdf and https://journals.plos.org/plosone/s/file?id=ba62/PLOSOne_formatting_sample_title_authors_affiliations.pdf 2. Thank you for stating the following in the Acknowledgments Section of your manuscript: The authors express their gratitude to all participants involved in this study and acknowledge the laboratory staff of the Buea Regional Hospital for their technical assistance; they also appreciate the support from the trimester research modernization funding provided by the Ministry of Higher Education of Cameroon to Dr. Arnaud Fondjo Kouam. We note that you have provided funding information that is not currently declared in your Funding Statement. However, funding information should not appear in the Acknowledgments section or other areas of your manuscript. We will only publish funding information present in the Funding Statement section of the online submission form. Please remove any funding-related text from the manuscript and let us know how you would like to update your Funding Statement. Currently, your Funding Statement reads as follows: The author(s) received no specific funding for this work.  Please include your amended statements within your cover letter; we will change the online submission form on your behalf. 3. We note that this data set consists of interview transcripts. Can you please confirm that all participants gave consent for interview transcript to be published? If they DID provide consent for these transcripts to be published, please also confirm that the transcripts do not contain any potentially identifying information (or let us know if the participants consented to having their personal details published and made publicly available). We consider the following details to be identifying information:- Names, nicknames, and initials- Age more specific than round numbers- GPS coordinates, physical addresses, IP addresses, email addresses- Information in small sample sizes (e.g. 40 students from X class in X year at X university)- Specific dates (e.g. visit dates, interview dates)- ID numbers Or, if the participants DID NOT provide consent for these transcripts to be published:- Provide a de-identified version of the data or excerpts of interview responses- Provide information regarding how these transcripts can be accessed by researchers who meet the criteria for access to confidential data, including:a) the grounds for restrictionb) the name of the ethics committee, Institutional Review Board, or third-party organization that is imposing sharing restrictions on the datac) a non-author, institutional point of contact that is able to field data access queries, in the interest of maintaining long-term data accessibility.d) Any relevant data set names, URLs, DOIs, etc. that an independent researcher would need in order to request your minimal data set. For further information on sharing data that contains sensitive participant information, please see: https://journals.plos.org/plosone/s/data-availability#loc-human-research-participant-data-and-other-sensitive-data If there are ethical, legal, or third-party restrictions upon your dataset, you must provide all of the following details (https://journals.plos.org/plosone/s/data-availability#loc-acceptable-data-access-restrictions):a) A complete description of the datasetb) The nature of the restrictions upon the data (ethical, legal, or owned by a third party) and the reasoning behind themc) The full name of the body imposing the restrictions upon your dataset (ethics committee, institution, data access committee, etc)d) If the data are owned by a third party, confirmation of whether the authors received any special privileges in accessing the data that other researchers would not havee) Direct, non-author contact information (preferably email) for the body imposing the restrictions upon the data, to which data access requests can be sent?

Reviewers' comments:

Reviewer's Responses to Questions

**Comments to the Author**

1. Is the manuscript technically sound, and do the data support the conclusions?

Reviewer #1: Partly

2. Has the statistical analysis been performed appropriately and rigorously?

Reviewer #1: I Don't Know

3. Have the authors made all data underlying the findings in their manuscript fully available?

Reviewer #1: Yes

4. Is the manuscript presented in an intelligible fashion and written in standard English?

Reviewer #1: No

**Reviewer #1: ** The manuscript by Kouam et al., tried to examine the association of liver disease and other risk factors with T2D. I have the following comments that may improve this manuscript. This manuscript is fine, but in my opinion these suggestions will improve it.

-in abstract please remove the statement ''...................leading to complications, including liver damage in line 37 - 38. the liver damage is not from the established complications of T2DM (Dunya Tomic et al, The burden and risks of emerging complications of diabetes mellitus., Nat Rev Endocrinol. 2022 Jun 6;18(9):525–539. doi: 10.1038/s41574-022-00690-7. I acknowledge that liver disease was linked to DM but not one of the traditional complications. The authors should rectify or remove this statement.

-in introduction, the statement ...............insulin in target tissues such as muscles and the liver'' lines 71-72, the authors should include adipose among the insulin target tissues. See Petersen and Shulman, Mechanisms of Insulin Action and Insulin Resistance, Physiol Rev, Actions, 2018 Oct 1;98(4):2133-2223. doi: 10.1152/physrev.00063.2017

- in introduction, the statement line 81-84..................prompt medical intervention to prevent further detrimental complications for the patients'' the authors should cite this information. This statement is 4 line long please split.

-although the manuscript is not difficult to read language revision is required.

- in introduction the authors should mention the well known complications of diabetes mellitus.

- materials and methods, the authors should state the inclusion and exclusion criteria in this study.

- what are the comorbidities examine in this study. Please specify in details.

-materials and methods, the authors should mention in details the methodology how they measurement these enzyme kits used, what manufacturers?, spectrophotometer used ? manufacturers?. Expand lines 145 to 152 in more details such that the results are reproducible.

- in table 2 of patients clinical characteristics the authors should include the bilirubin values.

-in table2 what types of comorbidities you mean, please mention in details.

-Results - I suggest the authors make an addition figure explaining the complications (microvascular and macrovascular) of DM and propose where the liver is classified.

- in discussion, please rectify the statement ''.....................complications, including those affecting the liver'' as the liver diseases are not from the classical complications of T2DM.

-the authors should discuss, propose or hypothesize the underlying mechanisms by which T2DM influences hepatic tissue and functions .

-the authors should include the limitations of this study for example limited sample size or other liver function tests were not included in this study.

- in discussion, the statement 278-279, ''These included the duration of illness, practice of physical activity, tobacco smoking, presence of comorbidities'' the authors should provide examples of the comorbidities included in this study.

-in discussion, statement "Moreso, regular physical activity is universally recognized for its beneficial effects on metabolic health, especially in managing diabetes [20,21]'' please explain this sentence

**Do you want your identity to be public for this peer review?** For information about this choice, including consent withdrawal, please see our Privacy Policy

Reviewer #1: No

---

## [Author Response · Author response to Decision Letter 1]

25 Jun 2025

RESPONSE TO THE REVIEWER’S COMMENTS

Note: All changes or suggestions are written in red in the revised manuscript

Dear Editor-in-Chief of the journal: PLOS ONE

Thank you for your letter dated June 19th, 2025, and the opportunity given to us to revise and resubmit the manuscript entitled “Abnormal Serum Levels of Liver Enzyme Markers and Related Risk Factors in Type 2 Diabetes Mellitus Patients Attending the Buea Regional Hospital, Cameroon. PONE-D-25-20030”. We would also like to take this opportunity to thank the editorial team and reviewers for their helpful comments, which greatly contributed to improving the current version of this manuscript. The manuscript has been updated in accordance with the reviewers' recommendations. Most of their inquiries have been answered, and some clarifications have been provided. Throughout this revised manuscript, modifications are written in red.

Editor and Reviewer comments

Editor’s comment

Comment #1: When submitting your revision, we need you to address these additional requirements.

Please ensure that your manuscript meets PLOS ONE's style requirements, including those for file naming. The PLOS ONE style templates can be found at https://journals.plos.org/plosone/s/file?id=wjVg/PLOSOne_formatting_sample_main_body.pdf and

Author Response: Dear Editor, thank you for your remark. The name of all files associated to this submission has been revised according to PLOS ONE’s style requirements.

Comment #2: Thank you for stating the following in the Acknowledgments Section of your manuscript: The authors express their gratitude to all participants involved in this study and acknowledge the laboratory staff of the Buea Regional Hospital for their technical assistance; they also appreciate the support from the trimester research modernization funding provided by the Ministry of Higher Education of Cameroon to Dr. Arnaud Fondjo Kouam.

Author Response: Dear Editor, thank you for your suggestions. The funding source indicated here is not a specific research fund with a typical Grant Number. It is a modest financial support allocated to the Cameroonian’s University Lecturer to facilitate their research activities in the field. The funding related sentence has been deleted in the Acknowledgment Section in the revised manuscript and included in the Cover Letter as follows:

Funding: This work was supported by the Trimester Research Modernization Funding granted by the Ministry of Higher Education of Cameroon. The funder had no role in study design, data collection and analysis, decision to publish, or preparation of the manuscript.

Thank you very much for your assistance in updating the statement online on our behalf.

Comment #2: We note that this data set consists of interview transcripts. Can you please confirm that all participants gave consent for interview transcript to be published? If they DID provide consent for these transcripts to be published, please also confirm that the transcripts do not contain any potentially identifying information (or let us know if the participants consented to having their personal details published and made publicly available). We consider the following details to be identifying information:

- Names, nicknames, and initials

- Age more specific than round numbers

- GPS coordinates, physical addresses, IP addresses, email addresses

- Information in small sample sizes (e.g. 40 students from X class in X year at X university)

- Specific dates (e.g. visit dates, interview dates)

- ID numbers

Author Response: Dear Editor, thank you for your suggestions. We hereby confirm that all participants gave their consent for the interview transcripts to be published. We also confirm that the transcripts do not contain any potentially identifying information. Also, the spreadsheet (Additional information: S3_file) columns do not contain any personal information and the participants are indicated by codes.

Reviewer’s comment

Reviewer #1: The manuscript by Kouam et al., tried to examine the association of liver disease and other risk factors with T2D. I have the following comments that may improve this manuscript. This manuscript is fine, but in my opinion these suggestions will improve it.

Comment #1: In abstract please remove the statement ''...................leading to complications, including liver damage in line 37 - 38. The liver damage is not from the established complications of T2DM (Dunya Tomic et al, The burden and risks of emerging complications of diabetes mellitus., Nat Rev Endocrinol. 2022 Jun 6;18(9):525–539. doi: 10.1038/s41574-022-00690-7. I acknowledge that liver disease was linked to DM but not one of the traditional complications. The authors should rectify or remove this statement.

Author Response: Dear Reviewer, thank you for your remark. The statement has been edited in the revised manuscript as follows: “Besides the traditional complications associated with T2DM, such as diabetic retinopathy, neuropathy, and kidney diseases, new complications including liver diseases, are increasingly being documented”.

Comment #2: -in introduction, the statement ...............insulin in target tissues such as muscles and the liver'' lines 71-72, the authors should include adipose among the insulin target tissues. See Petersen and Shulman, Mechanisms of Insulin Action and Insulin Resistance, Physiol Rev, Actions, 2018 Oct 1;98(4):2133-2223. doi: 10.1152/physrev.00063.2017

Author Response: Dear Reviewer, thank you for your suggestion. It has been corrected in the revised manuscript.

Comment #3: in introduction, the statement line 81-84..................prompt medical intervention to prevent further detrimental complications for the patients'' the authors should cite this information. This statement is 4 line long please split.

Author Response: Dear Reviewer, thank you for your remark. In the revised manuscript, the statement has been split into two sentences, and back up with proper citation.

Comment #4: Although the manuscript is not difficult to read language revision is required.

Author Response: Dear Reviewer, thank you for your remark. The manuscript has been thoroughly reviewed, and all grammatical errors have been corrected. All changes are written in red in the revised manuscript.

Comment #5: In introduction, the authors should mention the well-known complications of diabetes mellitus.

Author Response: Dear Reviewer, thank you for your suggestion. It has been corrected in the revised manuscript, and proper citation has been added.

Comment #6: materials and methods, the authors should state the inclusion and exclusion criteria in this study.

Author Response: Dear Reviewer, thank you for your comments. The inclusion and exclusion criteria have been indicated in the revised manuscript. Please, refer to Page 4, Line 102-107.

Comment #7: what are the comorbidities examined in this study. Please specify in detail.

Author Response: Dear Reviewer, thank you for your comment. The comorbidities examined in this study were conditions such as High Blood Pressure, Kidney Diseases, Cardiovascular diseases, Diabetes neuropathy and retinopathy. This has been indicated in the additional information (Questionnaire), S2_file

Comment #8: materials and methods, the authors should mention in detail the methodology of how they measure these enzyme kits used, what manufacturer? spectrophotometer used? manufacturers? Expand lines 145 to 152 in more detail such that the results are reproducible.

Author Response: Dear Reviewer, thank you for your comment. In the revised manuscript, detailed procedure of the measurement of liver enzyme activities was added. Also, information regarding the assay kits, the semi-automated spectrophotometer used as well as their respective manufacturers were provided. Modifications are highlighted in red.

Comment #9: In table 2 of patients clinical characteristics the authors should include the bilirubin values.

Author Response: Dear Reviewer, thank you for your suggestion. We agree that it would be better to include the bilirubin values. Unfortunately, in this study, we did not measure bilirubin content. That’s why we could not at it. Thank you for your understanding.

Comment #10: In table2 what types of comorbidities you mean, please mention in detail.

Author Response: Dear Reviewer, thank you for your suggestion. The comorbidities examined in this study have been indicated in the footnote of Table 2.

Comment #11: Results - I suggest the authors make an addition figure explaining the complications (microvascular and macrovascular) of DM and propose where the liver is classified.

Author Response: Dear Reviewer, thank you for your suggestion. The literature has already reported the complications (microvascular and macrovascular) of diabetes mellitus (Dunya Tomic et al, The burden and risks of emerging complications of diabetes mellitus., Nat Rev Endocrinol. 2022 Jun 6;18(9):525–539. doi: 10.1038/s41574-022-00690-7). In this research, it is clearly explained that liver damage in diabetes isn't a classic microvascular complication (kidney diseases, retinopathy or neuropathy), a traditional macrovascular complication (cardiovascular diseases, stroke). Liver damage is instead classified among the emerging complications, and it can significantly impact overall health and increase the risk of both macrovascular and, potentially, microvascular issues. Given that this classification has already been done and that it is not a direct result of this study, we think that adding a new figure will not be appropriate. However, we have discussed about it the discussion of the revised manuscript.

Comment #12: in discussion, please rectify the statement ''.....................complications, including those affecting the liver'' as the liver diseases are not from the classical complications of T2DM.

Author Response: Dear Reviewer, thank you for your suggestion. The statement has been rectified. Please, refer to page 15, Line 303-305 of the revised manuscript.

Comment #13: the authors should discuss, propose or hypothesize the underlying mechanisms by which T2DM influences hepatic tissue and functions.

Author Response: Dear Reviewer, thank you for your comment. The mechanisms by which T2DM influences hepatic tissues and functions have been added in the revised manuscript, with proper citation. Please, refer to Page 16, Line 317-322.

Comment #14: The authors should include the limitations of this study for example limited sample size or other liver function tests were not included in this study.

Author Response: Dear Reviewer, thank you for your comment. The limitations have been included in the discussion section of the revised manuscript. Please, refer to Page 18, Line 380-387.

Comment #15: In discussion, the statement 278-279, ''These included the duration of illness, practice of physical activity, tobacco smoking, presence of comorbidities'' the authors should provide examples of the comorbidities included in this study.

Author Response: Dear Reviewer, thank you for your remarks. Examples of comorbidities examined in this study have been added. Please, refer to Page 16, Line 314-315.

Comment #9: In discussion, statement "Moreso, regular physical activity is universally recognized for its beneficial effects on metabolic health, especially in managing diabetes [20,21]'' please explain this sentence

Author Response: Dear Reviewer, thank you for your remarks. The sentence simply means that physical activity can improve the metabolic function, and subsequently the overall health condition of diabetic patients, given that diabetes is a metabolic disease. In the revised manuscript, “metabolic health” has been replaced by “metabolic function”.

The authors are grateful to the reviewers for their valuable contributions which significantly improved the quality of this work. We very much hope the revised manuscript is accepted for publication in your Journal. Thank you for your consideration.

Sincerely yours,

Corresponding author

---

## [Decision Letter · Decision Letter 1]

Abnormal Serum Levels of Liver Enzyme Markers and Related Risk Factors in Type 2 Diabetes Mellitus Patients Attending the Buea Regional Hospital, Cameroon

PONE-D-25-20030R1

Dear Dr. KOUAM,

We’re pleased to inform you that your manuscript has been judged scientifically suitable for publication and will be formally accepted for publication once it meets all outstanding technical requirements.

Kind regards,

Suyan Tian

Academic Editor

PLOS ONE

Additional Editor Comments (optional): All points raised by the reviewer have been addressed. 

Reviewers' comments:

Reviewer's Responses to Questions

**Comments to the Author**

Reviewer #1: All comments have been addressed

2. Is the manuscript technically sound, and do the data support the conclusions?

Reviewer #1: Yes

3. Has the statistical analysis been performed appropriately and rigorously?

Reviewer #1: Yes

4. Have the authors made all data underlying the findings in their manuscript fully available?

Reviewer #1: Yes

5. Is the manuscript presented in an intelligible fashion and written in standard English?

Reviewer #1: Yes

Reviewer #1: (No Response)

**Do you want your identity to be public for this peer review?** For information about this choice, including consent withdrawal, please see our Privacy Policy

Reviewer #1: **Yes: ** Imadeldin Elfaki

---

## [Editor Report · Acceptance letter]

PONE-D-25-20030R1

PLOS ONE

Dear Dr. Kouam,

I'm pleased to inform you that your manuscript has been deemed suitable for publication in PLOS ONE. Congratulations! Your manuscript is now being handed over to our production team.

Kind regards,

on behalf of

Dr. Suyan Tian

Academic Editor

PLOS ONE